# Reliability of the Modified Ashworth and Modified Tardieu Scales with Standardized Movement Speeds in Children with Spastic Cerebral Palsy

**DOI:** 10.3390/children9060827

**Published:** 2022-06-03

**Authors:** Myungeun Yoo, Jeong Hyeon Ahn, Dong-wook Rha, Eun Sook Park

**Affiliations:** Department and Research Institute of Rehabilitation Medicine, Severance Hospital, Yonsei University College of Medicine, Seoul 03722, Korea; yme8902@yuhs.ac (M.Y.); ajh1208@yuhs.ac (J.H.A.); medicus@yuhs.ac (D.-w.R.)

**Keywords:** cerebral palsy, spasticity, Modified Ashworth Scale, Modified Tardieu Scale, reliability

## Abstract

The Modified Ashworth Scale (MAS) and Modified Tardieu Scale (MTS) are widely used to quantify spasticity. However, the reliability of their use for ankle plantar flexors has been questioned. In this study, we aimed to examine whether their reliabilities could be increased to acceptable levels for ankle plantar flexors using standardized movement speed in children with spastic cerebral palsy. The MAS and MTS scores for 92 limbs were assessed by two raters on two occasions, 1 week apart. A metronome was used to maintain the stretching velocity at 120 beats per minute. The intraclass correlation coefficients (ICCs) of the intra-rater reliabilities of the MAS and MTS and inter-rater reliability of the MAS were over 0.7. However, the ICCs for the inter-rater reliability of the MTS were <0.7 and >0.75 for the gastrocnemius and soleus muscles, respectively. The ICCs for the inter- and intra-rater reliabilities of the R1 angles ranged from 0.68 to 0.84, while those of the R2 angles ranged from 0.74 to 0.93. The reliabilities of the R2-R1 angles were not satisfactory. In conclusion, with a standardized movement speed, the reliability of the MAS for the ankle plantar flexors and the MTS for the soleus were satisfactory; however, that of the MTS for the gastrocnemius was not.

## 1. Introduction

Cerebral palsy (CP) is the most common childhood movement disorder. Spastic CP, the most common type, occurs in 78.2–86% of children with CP [1]. Botulinum toxin (BoNT) injections have been widely used for the management of spasticity in children with CP, and the ankle plantar flexor is the most common target muscle [2,3]. The accurate quantification of plantar flexor spasticity is important in determining the appropriate therapeutic strategies and documenting post-interventional changes.

The Modified Ashworth Scale (MAS) is the most widely used clinical tool for the measurement of spasticity. However, its reliability has been questioned in some studies [4,5,6]. In children with spastic CP or hypertonia, the reliability of MAS varied across studies [7,8,9,10]. In addition, the reliability differed considerably depending on the muscle tested, with the ankle plantar flexor having the lowest or second lowest reliability in children with spastic CP [7,8,9,10,11].

The Modified Tardieu Scale (MTS) is another tool commonly used to quantify spasticity. However, its reliability also varied across studies and depended on the muscle tested [7,9,10]. Specifically, the ankle plantar flexor MTS exhibited only low-to-moderate reliability in children with spastic CP [8,9,10,11].

Previous studies have highlighted the importance of the moving joint stretching velocity for testing reliability [12,13,14]. Marinelli et al. showed that clinical assessments using an external pacing device, such as a metronome, can be easily performed and achieve discrete velocities during linear passive movements in healthy subjects and stroke patients [15]. Metronomes have been used to standardize the stretch velocity during MAS-based assessments in adults with upper motor neuron lesions [16]. However, improving the reliability of these clinical tools with standardized movement speeds using a metronome has not been attempted in children with spastic CP. Therefore, the aim of this study was to examine whether assessing MAS and MTS scores using a metronome increases their reliabilities to acceptable levels for ankle plantar flexors in children with spastic CP.

## 2. Materials and Methods

### 2.1. Study Design

This was a prospective cross-sectional observational study. This study received ethical approval from the Institutional Review Board (IRB) and the Ethics Committee of Severance Hospital (#4-2020-0568). Informed consent for participation was obtained from the parents of all children participating in this study according to the IRB rules of our hospital. In addition, oral or written assent was also obtained from children over 7 years old, according to their understanding and cognitive abilities.

### 2.2. Participants

This study was conducted in a rehabilitation hospital affiliated with a university. Children admitted to Severance Hospital for intensive therapy between May 2021 and November 2021 were consecutively screened. The inclusion criteria were as follows: (1) diagnosis of spastic CP with ankle hypertonia; (2) age ranging from 1 to 15 years; (3) ability to cooperate with the study protocol; and (4) absence of contraindications for joint movements. The exclusion criteria were as follows: (1) change in medications or other interventions for spasticity between testing sessions; (2) previous botulinum toxin injections or short leg cast within 3 months; (3) history of a selective posterior rhizotomy or intrathecal baclofen pump; (4) maximal ankle joint dorsiflexion< −10 degree; (5) use of oral muscle relaxants; (6) severe or profound intellectual disability; (7) other types of cerebral palsy (such as dyskinetic, ataxic, and mixed type); and (8) lack of parental consent. As a result, 48 children (44, bilateral CP; 4, unilateral CP; 23 boys, 25 girls) were recruited. The mean age of the children was 78.1 ± 43.7 months (range: 15 months to 15 year 10 months). The Gross Motor Function Classification System levels ranged from I to V; patients were classified as levels I (10), II (9), III (15), IV (10), and V (4), respectively. The right and left limbs in bilateral CP and the affected limb in unilateral CP were assessed; thus, data for 92 total limbs were analyzed.

### 2.3. Measurements

The MAS, MTS, and R1 (angle of muscle reaction) and R2 (angle of full range of motion) angles in MTS were assessed by two raters on two occasions, 1 week apart. All tests were performed in the supine position with a midline head position. A neutral hindfoot position was maintained to avoid calcaneal valgus or varus. For the ankle plantar flexors with the knee extended, the hip and knee joints were at maximal extension, and the foot was moved from maximal plantarflexion to the maximal possible dorsiflexion. In this study, hypertonia from the ankle plantar flexors with the knee extended was assessed using only the gastrocnemius (GCM), rather than the GCM with the soleus. Then, the hip and knee joints were flexed 90° for the ankle plantar flexors with the knee flexed (soleus). To control the velocity of joint stretching, the metronome was set to 120 beats per minute (BPM), and the joint was stretched fully in 0.5 s to the rhythmic signal provided by the metronome for the fast stretching of the MTS and MAS. Theoretically, to provoke the stretching reflex, the raters moved the limb segment at minimally 130° per second, which means the ankle plantar flexor should be moved within maximally 0.5 s for children [15,16]. The two raters, including a physiatrist (rater 1) and a nurse practitioner (rater 2), underwent a training and performed the evaluation. Raters held meetings to practice and clarify the procedure according to the written guidelines of the two scales. The raters focused on controlling the velocity of moving the ankle joint using Pro Metronome app (EUMLab, Xanin Technology GmbH, Berlin, Germany). After listening to 3–5 beats of the metronome, the rater moved the ankle joint following the pace of the metronome, which was set to 120 BPM for the fast velocity. After each procedure, the raters reviewed the results. A training period of 1 month was sufficient for the two raters to learn to use the MAS and MTS with a controlled stretch velocity using a metronome. The values of the mean speed of joint stretch and the percent deviation for 92 limbs were described in Table 1. The percent deviation, which confirmed the difference between the actual speed of the joint stretch and the standard speed (130°/s), was calculated as: percent deviation (%) = (the mean speed of joint stretch −130)/130 × 100.

For each child, the raters moved the limbs three times. During the last movement, the R2 and R1 angles, MTS score, and MAS score were measured (Figure 1). First, a rater stretched the joint ‘as slowly as possible’ through the full range of motion to measure the R2 angle. Afterwards, the joint was moved by the rater ‘as quickly as possible’ in the same direction to the spasticity-provoked point of ‘catch’ of the third quick stretch to measure the R1 angle. Then, the quality of muscle reaction assessed using the MTS was graded from 0 to 4 [17]. The MAS was used last. The resting period between assessments was set to 5 min in an attempt to standardize resting for back to baseline [7]. For bilateral CP, the assessment sequence for the right and left sides was randomly allocated in a 1:1 ratio using a random number generator in Microsoft Excel software. All assessments were performed while the child was emotionally stable and without anxiety or fear. The raters performed the assessments independently, and the results were recorded separately for each rater and assessment. The data from the two different raters were compiled and analyzed by a physiatrist blinded to the study.

The measures from trials 1 and 2 for the same patient for each rater were compared to determine intra-rater reliability. Trials 1 and 2 for the same patients for each rater was used to assess interrater reliability. The intra and interrater reliabilities were analyzed with Intraclass correlation coefficients (ICCs) using SPSS version 25.0 (IBM Corp, Armonk, NY, USA).

The SEM is a measure of the precision of an assessment tool [18,19], which was calculated as: SEM = SD√(1−r), where SD is the standard deviation, and r is the reliability coefficient of the first of the test-retest assessments [20]. The SEM of the MAS and MTS was calculated.

### 2.4. Statistical Analysis

The ICCs can be used for ordinal data with equidistant intervals [21]. Based on the study by Pandyan et al. [12], the MAS and MTS scores were considered ordinal, and an MAS score of 1^+^ was assigned to ratings of 1.5 to maintain equal intervals. ICC was computed to assess the intra- and inter-rater reliabilities of the MTS, MAS, R1, R2, and R2-R1 scores. A two-way mixed-consistency ICC model was assessed using SPSS version 25.0 (IBM Corp, Armonk, NY, USA). Using the standards suggested by Portney and Watkins [21], the clinical significance was defined as low for an ICC of <0.50, moderate for 0.50–0.75, and good for ≥0.75.

## 3. Results

The ICC values for the intra- and inter-rater reliabilities of the MAS ranged from 0.70 to 0.88. Its intra-rater reliability was good for both the GCM and soleus muscles. The ICC scores for the inter-rater reliability of the MAS was good only for the soleus but not for the GCM. The ICC scores for the intra-rater MTS reliability ranged from 0.70 to 0.78. The intra- and inter-rater reliabilities of the MTS were good for the soleus but not for the GCM. The ICC scores for intra-rater R1 reliability ranged from 0.76 to 0.84. The intra-rater R1 reliability was good for both the GCM and soleus muscles. In contrast, the inter-rater R1 reliability was moderate to good, with ICC scores ranging from 0.68 to 0.80. In addition, the ICC scores for intra- and inter-rater R2 reliabilities ranged from 0.74 to 0.93. Seven of eight R2 angle measurements had good reliabilities. In contrast, the R2-R1 angle exhibited moderate reliability, with ICC scores ranging from 0.54 to 0.74 (Table 2 and Table 3). The SEMs are described in Table 4.

## 4. Discussion

The MAS does not measure the velocity-dependent aspect of spasticity; thus, the validity of the MAS for quantifying spasticity has been questioned [4,5,6]. Despite this limitation, the MAS is the most widely used tool due to its ease of application, short testing time, and lack of requirements for specialized equipment [10]. However, the reliability of the MAS in children with CP has not yielded satisfactory results in most studies. A lack of standardization of the stretch velocity is one of the factors related to the unsatisfactory reliability of the MAS. According to a previous study, standardizing the movement speed with a metronome helped achieve a reasonable inter-rater reliability; however, the reliability was reasonable for the elbow flexor but not for the ankle plantar flexor [16]. According to previous studies [22,23,24], the MAS exhibited better reliability for the upper limbs than for the lower extremities. One possible explanation is that the lower limbs have greater lengths and overall muscle masses than the upper limbs, which makes them weigh more. The weight of the limbs may affect reliability [12,22]. Thus, children have lighter lower extremities, which can make the handling and testing easier; therefore, the reliability of the MAS in children is likely to be higher than that in adults. In contrast, the characteristics of children, such as being easily agitated and restless, may adversely affect the reliability. The difficulties of assessing measurement tools for children with CP have been highlighted by Fosang et al., who demonstrated a wide variability in the MTS reliability for the test–retest measurements of six raters [7].

The limited number of studies on the MAS reliability in children with CP have demonstrated low-to-moderate MAS reliabilities [7,8,9,10,11]. The ICC scores for inter- and intra-rater reliabilities ranged from 0.21 to 0.72 for the GCM and from 0.33 to 0.54 for the soleus. In comparison, the ICC scores for reliability for MAS in the present study were higher. In a study by Fosang et al. [7], an ICC of 0.7 was considered an acceptable limit for reliability. Accordingly, the inter- and intra-rater reliabilities of MAS in the present study were acceptable. There are many factors related to the wide variability in reliabilities across studies, such as sample size, muscle tested, weight of the tested limb, number of raters, test position, number of repetitions, and lack of standardized assessment methods [22]. A standardized movement speed using a metronome is one method to enhance reliability. The overall higher ICC scores in the present study support the importance of the moving joint stretching velocity for reliability testing [12,13,14].

Fosang et al. found that the inter-rater reliability of the MTS and R2 (passive range of motion) was acceptable, with an ICC of 0.7 [7]. In contrast, other studies found the intra- and inter-rater MTS reliabilities were low-to-moderate, with an ICC ranging from 0.22 to 0.63 [9,10]. In the present study, the ICC scores for intra- and inter-rater reliabilities of the MTS were >0.7, which were higher than the results of previous studies, except the inter-rater reliability of the MTS for the GCM. In the present study, the intra- and inter-rater reliabilities were greater for the soleus than for the GCM. The MTS intra-rater reliability was greater for the GCM than for the soleus in a study by Numanoglu et al., whereas its inter-rater reliability was greater for the soleus than for the GCM in a study by Yam et al. [9,10]. The inconsistent results of the MTS reliability between the GCM and soleus should be delineated in further studies. The SEM is an estimate of the error value associated with measurement. The smaller the value, the more accurate the measurement. The SEM values of the MAS and MTS using a metronome were found to be low in present study, which supports the test and retest reliability of the method using a metronome.

To the best of our knowledge, there are only two studies on the reliability of the R1 and R2 angles of MTS in children with CP. The intra- and inter-rater reliabilities of the R2 and R1 angles were good in the study by Numanoglu et al. and were low to moderate in the study by Yam et al. [9,10]. The present study demonstrated moderate-to-good intra- and inter-rater reliabilities for the R1 and R2 angles. The study by Yam et al. had a much smaller sample size than our study and that of Numanoglu et al., which may explain the low reliability in their study. For measuring the R1 and R2 angles, the speed of joint movement was standardized to be “as quickly as possible” or “as slowly as possible.” The R2 angles exhibited a greater reliability than the R1 angles, both in the present study and compared with in previous studies.

The difference between the R2 and R1 angles indicates dynamic spasticity, and greater angular differences may benefit from BoNT-A injections. The study by Numanoglu et al. demonstrated moderate-to-good intra-rater reliabilities of the R2-R1 angle [10]; whereas, the study by Yam et al. demonstrated low-to-moderate inter-rater reliabilities of the R2-R1 angles [9]. In the present study, the inter- and intra-rater reliabilities were moderate. According to the study by Mackey et al. [25], the R2-R1 angles showed a large intersessional variability for elbow flexor muscles in 10 children with hemiplegia CP; thus, they concluded that the R2-R1 angle may have a limited value for assessing the biceps spasticity in children with CP. The results of the present study also support the limited value of the R2-R1 angles for assessing the dynamic ankle plantar flexor spasticity as a research tool in these children.

A limitation of this study was that only the ankle plantar flexors were assessed. Although ankle plantar flexors are the most frequent target muscles for BoNT-A injections in children with CP, the ankle plantar flexors had the lowest MTS and MAS reliabilities among the muscles tested. In addition, children often could not tolerate the prolonged evaluation times; thus, only ankle plantar flexors were assessed in this study. However, the reliabilities may vary depending on the muscles tested. The usefulness of the metronome for enhancing the MAS and MTS reliabilities should be examined in other muscles in further studies. Despite this limitation, the study has significant value because of its large sample size compared with previous studies. Another limitation of the present study was that only two raters were assessed for examining reliabilities. According to a systemic review, the larger the number of raters, the lower the reliability [22]. Therefore, it is possible that the reliabilities would be lower if more raters were assessed. To document spasticity, examine the effect of therapeutic interventions, and minimize the impact of the last limitation, it is recommended that MAS and MTS be assessed by a single rater for the same patient.

## 5. Conclusions

The present study supports that standardization of stretch movement speed using a metronome is helpful for enhancing the reliabilities of the MAS and MTS for ankle plantar flexors in children with spastic CP. However, the usefulness should be examined in other muscles in these children. Without using a metronome, the reliabilities of the R1 and R2 angles ranged from moderate to good. On the other hand, the reliability of the R2-R1 angles were only moderate. The use of the R2-R1 angle to quantify the dynamic spasticity and measure outcomes after therapeutic interventions should be cautiously considered.

## Figures and Tables

**Figure 1 children-09-00827-f001:**
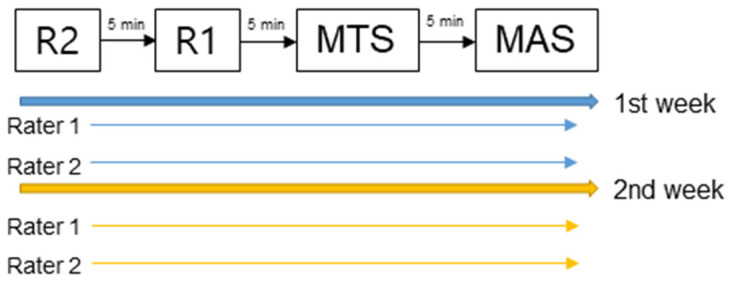
Flow diagram of the procedure.

**Table 1 children-09-00827-t001:** The actual mean speed of joint stretch and the percent deviation of the speed.

		Rater 1	Rater 2
First measurement	Knee flexion	131°/s (0.7%)	136°/s (4.6%)
	Knee extension	124°/s (−4.6%)	123°/s (−5.3%)
Second measurement	Knee flexion	132°/s (1.5%)	136°/s (4.6%)
	Knee extension	127°/s (−2.3%)	126°/s (−3.1%)

**Table 2 children-09-00827-t002:** Inter-rater reliability of the Modified Ashworth Scale and Modified Tardieu Scale for the first and second measurements.

Muscle Group	First Measurement					Second Measurement				
	MAS	MTS	R1	R2	R2-R1	MAS	MTS	R1	R2	R2-R1
Ankle PF, Knee extended	0.70 (0.55–0.80)	0.68 (0.52–0.79)	0.73 (0.59–0.82)	0.83 (0.74–0.89)	0.54 (0.30–0.70)	0.73 (0.58–0.82)	0.58 (0.34–0.72)	0.80 (0.70–0.87)	0.88 (0.82–0.92)	0.65 (0.47–0.77)
Ankle PF, Knee flexed	0.77 (0.65–0.84)	0.82 (0.73–0.88)	0.69 (0.51–0.80)	0.88 (0.80–0.92)	0.56 (0.34–0.71)	0.82 (0.71–0.89)	0.81 (0.71–0.87)	0.68 (0.51–0.79)	0.93 (0.88–0.95)	0.66 (0.48–0.77)

The values are presented as intraclass correlation (ICC) (95% confidence interval). *p*-values of <0.05 indicate statistical significance PF, plantar flexors; MAS, Modified Ashworth Scale; MTS, Modified Tardieu Scale; R1, angle of the muscle reaction; R2, angle of the full range of motion.

**Table 3 children-09-00827-t003:** Intra-rater reliability of the Modified Ashworth Scale and Modified Tardieu Scale.

Muscle Group	Rater 1					Rater 2				
	MAS	MTS	R1	R2	R2-R1	MAS	MTS	R1	R2	R2-R1
Ankle PF, Knee extended	0.76 (0.64–0.84)	0.70 (0.54–0.80)	0.84 (0.76–0.90)	0.74 (0.60–0.83)	0.68 (0.51–0.79)	0.83 (0.74–0.89)	0.74 (0.61–0.83)	0.78 (0.67–0.85)	0.84 (0.76–0.89)	0.62 (0.42–0.75)
Ankle PF, Knee flexed	0.83 (0.75–0.89)	0.76 (0.63–0.84)	0.76 (0.64–0.84)	0.84 (0.76–0.90)	0.71 (0.57–0.81)	0.88 (0.82–0.92)	0.78 (0.67–0.85)	0.82 (0.73–0.88)	0.90 (0.84–0.93)	0.74 (0.61–0.83)

The values are presented as intraclass correlation (ICC) (95% confidence interval). *p*-values of <0.05 indicate statistical significance. PF, plantar flexors; MAS, Modified Ashworth Scale; MTS, Modified Tardieu Scale; R1, angle of the muscle reaction; R2, angle of the full range of motion.

**Table 4 children-09-00827-t004:** Standard error of measurement for the Modified Ashworth Scale and Modified Tardieu Scale.

	MAS	MTS	R1	R2	R2-R1
Ankle PF, Knee extended	0.31	0.35	4.71	4.25	6.18
Ankle PF, Knee flexed	0.32	0.33	6.53	3.80	6.77

The values are presented as standard error of measurement (SEM). PF, plantar flexors; MAS, Modified Ashworth Scale; MTS, Modified Tardieu Scale; R1, angle of the muscle reaction; R2, angle of the full range of motion.

## Data Availability

The data presented in this study are available on request from the corresponding author. The data are not publicly available due to reasons concerning privacy of the participants.

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
