# Peer review of "Reliability of the Modified Ashworth and Modified Tardieu Scales with Standardized Movement Speeds in Children with Spastic Cerebral Palsy"

_children, 2022, doi:10.3390/children9060827_

Round 1

Reviewer 1 Report

This manuscript describes the investigation of intra and inter rater reliability of the Modified Ashworth Scale and Modified Tardieu Scale using a metronome to control speed of rater movement when assessing plantar flexors. Overall, the manuscript is easy to read and provides valuable information regarding the reliability of tone assessment in children with cerebral palsy. My recommendations for revision follow.

Line 70: Define severe in ‘severe fixed contracture’. Was there a cut off related to degrees of available movement?

Line 71: Recommend changing ‘severe mental retardation’ to severe or profound intellectual disability.

Section 2.2 Add data related to mixed tone vs pure spasticity of the cohort that participated in the study.

Line 89 states that the stretch for both the MAS and MTS were completed at a speed of .5 seconds. The MAS is described in the literature to be performed at a speed of one second and the MTS speed is described as less than one second. Provide more information on why .5 seconds was selected for both measures.

Line 90 Provide more information on the training period for the raters. What was included in the training? How were procedures standardized?

Line 230 Change exam to examine.

Section 4 Discussion – Based on your experience with the study, do you have additional recommendations to improve reliability of tone assessment in children with CP that could be investigated in future studies?

Reviewer 2 Report

see attached
